# Clinical Evaluation of Nerve Function in Electrical Accident Survivors with Persisting Neurosensory Symptoms

**DOI:** 10.3390/brainsci12101301

**Published:** 2022-09-27

**Authors:** Andrew Wold, Lisa Rådman, Kerstin Norman, Håkan Olausson, Magnus Thordstein

**Affiliations:** 1University Health Care Research Centre, Faculty of Medicine and Health, Örebro University, 70182 Örebro, Sweden; 2Center for Social and Affective Neuroscience, Department of Biomedical and Clinical Sciences, Linköping University, 58185 Linköping, Sweden; 3Department of Physiotherapy, Faculty of Medicine and Health, Örebro University, 70182 Örebro, Sweden; 4National Unit for Health and Safety, Swedish Police Authority, 10226 Stockholm, Sweden; 5Neuro, Biomedical and Clinical Sciences, Linköping University, 58185 Linköping, Sweden; 6Department of Clinical Neurophysiology, Region Östergötland University Hospital, 58185 Linköping, Sweden

**Keywords:** neurophysiology, electrical accidents, neurography, QST, LEP, self-reported symptoms

## Abstract

**Highlights:**

**What are the main findings?**
A majority of electrical accident victims with persisting symptoms show small/large nerve fiber dysfunction.Abnormal QST and/or neurography results were present in 67% of patients.No test result strongly correlated with self-reported symptoms.

**Abstract:**

Objective: Work related electrical accidents are prevalent and can cause persisting symptoms. We used clinical neurophysiological techniques to assess neurosensory function following electrical accidents and correlated test results with the patients’ symptoms. Methods: We studied 24 patients who reported persisting neurosensory symptoms following a workplace electrical accident. We assessed nerve function using quantitative sensory testing (QST), thermal roller testing, laser evoked potential (LEP), and electroneurography. The patients’ results were compared with previously established normative data. Results: Altogether, 67% of the patients showed at least one neurosensory impairment with a large heterogeneity in test results across patients. At a group level, we observed significant deviations in in QST, LEP, and sensory and motor neurography. Overall, we found a weak correlation between test results and self-reported symptoms. Conclusions: In a majority of patients with neurosensory symptoms after a workplace electrical accident, neurosensory testing confirmed the existence of an underlying impairment of the nervous system.

## 1. Introduction

Millions of workers, in virtually all kinds of professions—not only electricians [1], are exposed to electricity in their daily workplace and are thereby at risk of suffering an electrical accident. In 2006, the British Health and Safety Executive reported that between 1996 and 2003, 5% of all industrial accidents involved contact with electricity, and that 10% of these were fatal [2]. Both industrial and home accidents account for up to approximately 30% of admissions to burns units in industrial countries [3]. Such accidents are 25 times more lethal than accidents due to falls, with the majority caused by human error [4]. Typical injuries due to high-powered electrical currents (>1000 V) passing through the body involve burns, superficial wounds, and lesions on the hands, head or upper limbs [5].

While accidents involving electricity are common, we still lack knowledge regarding their long-term consequences. The severity of accidents can range from minor to life-threatening, but many of these accidents carry less apparent long-term consequences [6,7]. If an accident is severe, the standard of immediate care can vary from treating superficial burns and running blood tests, to checking for cardiac arrhythmia [8]. However, only when patients complain about extreme symptoms, they may be referred to clinical neurophysiological examinations, such as neurography [3]. This contributes to an unclear picture of the neurological consequences of an electrical accident [9]. However, studies have evaluated electrical accidents through self-reporting and found that those affected experienced pain, muscle weakness, loss of sensation or reduced sensitivity and that these neurological symptoms continued for at least a year [10,11,12].

Considering the scope of neurological consequences following electrical accidents [12], the mechanisms behind these symptoms need to be investigated in more detail [12]. Our focus is on the peripheral nerve function, although central nervous function is known to have substantial alterations following electrical accidents. Nerve injuries may be caused by direct nerve destruction or nerve compression [13]. There is some evidence that electrical damage primarily causes axonal injury and not demyelination, [14,15], and there is a need to explore tests for evaluating patients being treated for electrical accidents. We have previously demonstrated reduced somatosensory abilities in people suffering an electrical accident: reduced hand thermal sensitivity to both heat and cold while vibration perception was within the normal range [1]. This suggests injury to the thinly myelinated and unmyelinated afferents responsible for the transmission of temperature alterations [15,16]. Here, we extended previous findings using several methods to test nerve fiber functionality in people reporting symptoms after suffering an electrical accident in the past two to six years. Using neurography, quantitative sensory testing (QST), thermal sensing and laser-evoked potentials (LEP), we characterized nerve fiber dysfunctions related to electrical accidents. We employed methods used in clinical practice and their normative data to investigate the relations between these results and the patients’ persisting symptoms, which are symptoms present more than one-year after an accident. Our aims are to test if clinical routines are able to detect nerve function abnormalities within a highly selected group of electrical accident patients with persisting symptoms related to their accident, and, if such abnormalities can be found, to establish which persisting symptoms best predict subsequent nerve function.

Knowledge is scarce regarding peripheral nerve function after electrical accidents [17]. We hypothesized that in a carefully selected group of patients displaying adequate long-term symptoms will show at least one abnormal neurophysiological test value. Treating acute electrical accidents focuses on immediate care, while neurophysiological symptoms often appear weeks to months after an accident [18]. Therefore, we only included a group selected for symptoms that have persisted for at least a year, and, to increase the chance of having an abnormal test result, only include those with symptoms of a moderate to considerable severity level. We also hypothesized that there would be a link between the tests with abnormal results and the persisting symptoms that resulted from an electrical accident.

## 2. Materials and Methods

### 2.1. Study Design and Study Population

We examined 24 patients (50% of whom were electricians) with persisting sensory symptoms following a work-related electrical accident (Figure 1). Using reports from the AFA insurance company (Stockholm, Sweden) between 2014 and 2019, we identified 1056 victims of electrical accidents in Sweden who suffered an electrical accident between one and six years ago (median three years). The AFA insurance company first asked for willingness to participate in the study. Only those who agreed went to the next step: a full questionnaire e-mail via Smart-trial© (Copenhagen, Denmark). Across Sweden, 160 individuals were sent the full questionnaire and in cases of no response, received a reminder after two weeks.

A total of 129 individuals (81%) responded to the initial questionnaire. In addition to having had an electrical accident, we needed to ensure that participants had persisting symptoms that pertained to their accident and not to another source. Our main inclusion criteria utilized two tests to verify electrical accident persisting symptoms: Self-Administrated Leeds Assessment of Neuropathic Symptoms and Signs (SLANSS) and the hand-arm vibration syndrome (HAVS) scale. SLANSS is a validated 11-item scale used to uncover the presence of neuropathic pain [19]. Participants can have a max score of 24, and a score of 12 or above indicates a presence of moderate to considerable symptoms. The HAVS is a validated questionnaire [10,16] that contains 10 items about hand symptoms with the scaled options of no, light, moderate and considerable. To be included, participants had to have a SLANSS score of ≥12 and report moderate or considerable symptoms on HAVS.

Exclusion criteria were defined to exclude individuals with other possible causes of affected nerve and/or hand function. Specifically, patients with Smart-trial responses indicating, vibration-induced numbness, muscle weakness, and finger whitening, earlier nerve injury—such as carpal tunnel syndrome or previous accidents, neurological diseases, diabetes, cancer treatment, or known vitamin deficiency were excluded. Altogether, 29 patients fulfilled the inclusion and met no exclusion criteria. Of these, 24 completed the entire investigation.

### 2.2. Procedure

The study was approved by the Swedish Ethical Review authority (2019-02102) and was conducted in accordance with the Declaration of Helsinki. The patients gave written consent before entering the study. Patients received financial compensation.

#### 2.2.1. Questionnaire

The web-based questionnaire was administered by Smart-trial. In addition to questions regarding inclusion and exclusion criteria, we collected data about the circumstances, the severity of the accident and symptoms.

#### 2.2.2. Examination

The 24 patients were examined at the Department of Clinical Neurophysiology, Linköping University Hospital, Sweden. All examinations were performed by one of the authors (A.W.).

#### 2.2.3. Means of Assessment

##### Quantitative Sensory Testing

Thermal perception thresholds for warm and cold were assessed using quantitative sensory testing (QST; Medoc TSA II, Ramat Yishai, Israel); a psychophysical test dependent on Aδ and C fiber function [20].

Thermal perception thresholds were assessed bilaterally on the thenar surface of the hand and the plantar surface of the foot arc. Skin temperature was measured before starting and, when required, the skin was warmed with a heated gel pad producing a starting temperature above 28 °C. The thermode (25 mm × 50 mm) was placed on the skin and the patients were instructed to “press the button when you identify a change in temperature”. The baseline temperature was set to 32 °C with a temperature change of 1 °C per second; six cold and warm stimulations were presented in a randomized order. From these six cold and warm stimulations, we calculated the mean and standard deviation of the thresholds. Age- and sex-specific normative data from the manufacturer of the Medoc TSA II system were used as reference materials [21]. Values exceeding two standard deviations in the sub-normal direction from the reference material (see Table 1) were marked as clinically relevant (see Appendix A). For safety reasons, the probe’s maximum temperature was set to 50 °C, and the minimum to 5 °C.

##### Temperature Roller Testing

Thermal perception was also examined using temperature-controlled metal rolls (Thermaroll, Somedic, Sösdala, Sweden), one for cold, at 25 °C, and one for warm, at 40 °C [22]. Patients received a cold and warm stimulation to each palm and foot sole in a randomized order, with approximately 15 s between exposures. Patients were asked if they felt a cold or warm sensation. The number of correct responses was recorded with a maximum possible score of eight (Appendix A). The test was only to be passed when patients were able to correctly respond to all stimuli delivered.

##### Laser-Evoked Potentials

Laser stimulation, which activates Aδ and C fibers [23], was applied to the dorsum of the hand and the foot and cortical evoked potentials were recorded with scalp electrodes. We used a neodymium: yttrium–aluminum–perovskite (Nd; YAP) laser (Stimul 1340, DEKA Ltd., Calenzano, Italy) set to 0.5 Hz, 10 ms pulse, and 4 mm spot size. To determine individual stimulus intensity, we started at 1.0 J intensity and increased stepwise by 0.5 J until patients reported pain intensity of four on a 10-point pain scale (where 0 indicated no felt sensation and 10 extreme pain). The stimulation never exceeded 2.5 J [24] to prevent dermal burns. To record potentials, we used commercial software (Curry 8, Compumedics Neuroscan, Abbotsford, Australia) with 12 standard surface electrodes using a 64-channel amplifier (SynAmps RT, Compumedics Neuroscan, Charlotte, NC, USA). We placed nine head electrodes corresponding to Fz, Cz, Pz, T3, T4, A1, A2, a ground electrode (between Cz and T4), and an ECG detection electrode (between Pz and T3). We also placed an electrode below each eye and one above the nasion, to facilitate the detection of blink artefacts. All channels had a maximum impedance of 10 μΩ and a band pass filter of 1 to 30 Hz. The area for the laser was shifted after each stimulation to prevent fatigue and burn lesions. The interstimulation interval varied from eight to twelve seconds. Each patient had both hands and feet stimulated with 20 and 30 stimulations, respectively. To determine the amplitude and latency for each hand or foot, we used a custom script (built in Python 3.8, Python, Amsterdam, The Netherlands) that allowed visual inspection of each stimulation. The LEP recordings were made using a standard technique, aiming to define N2 and P2 as the largest negative and positive peaks in the post-stimulus interval 0–500 ms [25]. Where we observed a clear artefact (such as a blink), we removed the stimulation from the total set. We used our own laboratory normal values (71 individuals, median age 30 [range: 19 to 60], 105 hand and 84 foot observations) for purposes of LEP evaluation and took the corrected average for max N2-P2 amplitude and shortest N2 latency [26] (see Table 2). Values exceeding two standard deviations in the sub-normal direction from our reference material were considered clinically relevant (see Appendix A).

##### Neurography

Damage of large diameter neurons may be detected as abnormal conduction parameters (latencies, velocities) and/or response amplitude. To investigate upper limb damage associated with electrical accidents, whilst avoiding the common confounder with undiagnosed carpal tunnel median nerve compression, we choose to focus on the motor and sensory function of the ulnar nerve. For the lower extremity, we choose the peroneal nerve for motor and the sural nerve for sensory parameters. Neurography was performed with a Dantec Keypoint system (Alpine Biomed ApS, Skovlunde, Denmark) in accordance with the Uppsala neurophysiology department guidelines (see Uppsala procedure reference) [25]. Skin temperature was measured before starting and, when required, the skin was warmed with a heated gel pad producing a starting temperature above 28 °C. After analysis of signal quality, we compared response latency, response amplitude, F-wave latency and conduction velocity against reference values (see Uppsala reference material) [27]. Customized reference values were computed using the Uppsala reference value constants and average age and height of our patient sample. Values exceeding two standard deviations in the sub-normal direction were marked as clinically relevant (see Appendix A).

##### Statistics for Tests and Symptoms

For comparing the group of electric accident victims with reference materials, we chose the Welch *t*-test that is well suited to situations when neither the group sizes nor the variances are equal. For the analysis, we used SPSS version 27.0 (IBM Corp., Armonk, NY, USA). To control for multiple comparisons, we corrected the alpha value with a Bonferroni correction. We considered the measurements to be independent of one another if they were separate tests (such as LEP, QST, neurography), performed on different nerves (ulnar nerve, peroneal nerve, sural nerve), or different sides of the body (left or right). We reported *p*-values as calculated, and used one (*p* < 0.05), two (*p* < 0.01), or three (*p* < 0.001) stars to indicate Bonferroni-corrected significances (see plot captions for details). We correlated data from the HAVS questionnaire with testing results, using Kendall tau-b as the correlation coefficient.

## 3. Results

### 3.1. Patients

Twenty-four patients underwent clinical testing (median age = 41.5 years, range 25–65; median height 177.5 cm, range 160–189; all right-handed; five female). All patients had workplace accidents. Twelve of the 24 were employed as electricians at the time of their accident. All but two were involved in low voltage electrical accidents (<1000 V), and all patients reported one of their hands as the electrical entry point. All but two patients sought medical care after their accidents, and they missed an average of 1.08 (SD = 1.35) days of work (for more demographic information, see Appendix A).

### 3.2. Quantitative Sensory Testing Results

Group QST values differed significantly from reference values for both cold (Figure 2) and warm (Figure 3) sensation on the hands and feet (Table 1). This was statistically significant at the corrected 0.01 *p*-value. Eight patients (33%) experienced at least one clinically relevant shift (>2 SD) in temperature perception, in either warm or cold sensations (median locations per affected patient = 3.5, range 1–8). Clinically relevant temperature sensitivity deviations were observed more often in hands (18/96 occurrences) than in the feet (10/96 occurrences), and were more often warm sensation deficits (25/96 occurrences) than cold sensation deficits (13/96 occurrences).

### 3.3. Temperature Roller Testing Results

For temperature roller testing, six of 24 patients (25%) had at least one mistaken or unidentifiable sensation. Of those six, five made a single incorrect assessment and one patient made four incorrect assessments. Comparing QST to the temperature rollers, four patients (29%) had overlapping QST and temperature roller instances. Of the nine instances of an incorrect temperature roller occurrence, four (44%) also had clinically relevant QST abnormalities.

### 3.4. Laser Evoked Potentials Results

Group LEP amplitudes were significantly reduced when compared to reference values (left foot, T_40_
_98_, *p* = 0.0006; right foot, T_38_
_49_, *p* ≤ 0.0005; right hand T_40_
_98_, *p* = 0.0061) (Table 2 and Figure 4); none were considered clinically relevant. LEP latencies only differed significantly for the right foot (T_40_
_98_, *p* = 0.0189; Table 3). One of the foot latencies (4%) were considered clinically relevant (<2 normal data SDs; Figure 5).

### 3.5. Neurography Results

Motor neurography latencies (Table 4 and Figure 6) and F-wave latencies (Table 5 and Figure 7) showed no differences compared to reference values. Although the group differences were not greater than reference values, we observed one clinically relevant left ulnar nerve motor and F-wave latency (within the same patient) and one increased F-wave latency in the right peroneal nerve of another patient (all 4%). Motor amplitudes on the left and right ulnar and peroneal nerve were significantly lower than reference values, although none were considered clinically relevant (Table 6 and Figure 8). Motor conduction velocities were significantly lower than the reference values for the left peroneal nerve (T_49_
_93_, *p* = 0.0061), and the left and right ulnar nerve below the elbow (left side, T_38_
_66_, *p* = 0.0008; right side, T_38_
_66_, *p* = 0.015; Table 7 and Figure 9). Four patients (17%) showed eight clinically significant reductions in conduction velocity. The abnormalities were in the left ulnar nerve distal to the elbow (*n* = 3), the left ulnar nerve proximal to the elbow (*n* = 2), the right ulnar nerve proximal to the elbow (*n* = 2), and in the right peroneal nerve (*n* = 1). Sensory neurography amplitudes showed no differences compared to reference material values (Table 8 and Figure 10). Although the group differences were not below reference values, we observed six clinically relevant instances of abnormality in five patients (21%) (all in the ulnar nerves [3 right and 3 left], Appendix A). Sensory conduction velocities showed differences compared to reference material values in the left sural nerve (T_32_
_33_, *p* = 0.0006), and bilaterally in the ulnar nerve (left side, T_40_
_59_, *p* ≤ 0.000167; right side, T_40_
_59_, *p* ≤ 0.000167; Table 9 and Figure 11). Six patients (25%) had reduced conduction velocity in 11 instances, with one patient accounting for five of those. Of the 11 instances, three were clinically relevant (left ulnar nerve palm, *n* = 4, right ulnar nerve palm, *n* = 2, left ulnar nerve ring finger, *n* = 1, right ulnar nerve ring finger, *n* = 2, and left ulnar nerve little finger, *n* = 1). Overall, eleven out of 24 patients (46%) showed at least one clinically relevant neurography abnormality (median instances per affected patient 2, range 1–7).

### 3.6. Correlation with Symptoms

In an exploratory analysis of symptoms, we ranked all symptoms based on severity/prevalence (see Figure 12) and correlated the self-reported symptom values with test values. All patients answered all of the correlated symptom questions. There is a clear lack of homogeneity when it comes to symptoms experienced by patients. Overall, there was weak to moderate correlation (corresponding to a Kendall tau-b values from ±0 to ±0.29), the vast majority of which were not statistically significant even before a post hoc correction. Both total clinically relevant cases for motor neurography and QST showed three strong correlations (>0.4), but none that were significant following a Bonferroni correction.

## 4. Discussion

We have demonstrated that otherwise healthy individuals with persisting symptoms following a workplace electrical accident showed altered nerve functionality when tested two to six years after the incident. We showed that this risk group extends beyond just electricians, as only half of our patients were electricians. We have replicated the findings regarding abnormalities in temperature sensitivity from our previous study [16] and have extended these findings to include clinical measurements of nerve fiber function. These revealed a mixed picture suggesting that motor nerves, as well as different types of large- and small-diameter sensory nerves, in the upper or lower extremity were differently affected in different patients. Cortical amplitudes measured using LEP were lower in both feet and the right hand. The LEP latency was significantly greater in the right foot only. Neurography measurements showed a decrease in motor amplitudes for the left and right peroneal nerve and ulnar nerves. Motor as well as sensory conduction velocities were significantly reduced for the left leg and left and right lower arms.

At the individual level, 16 out of 24 patients (67%) had one or more clinically relevant test result. This was from a group of carefully selected electrical accident sufferers with persisting symptoms, and a striking feature was the lack of a clear pattern among them. Eight out of 24 (33%) patients experienced a QST temperature sensitivity shift, while eleven out of 24 patients (46%) had a clinically relevant shift in performance for a neurography measurement. However, only three patients (13%) showed findings that significantly deviated in both of these tests. This was the case, even within the same domain (such as temperature sensitivity). The temperature roller testing and QST only overlapped for four patients (17%), and of the nine instances of incorrect temperature rolls, only four also had clinically relevant QST values. Thus, even when comparing two tests designed to test a similar function, the difference in test rigor produced different results.

The symptom pattern patient varied across patients and there were no corrected statistically significant correlations to individual test parameters. The correlation between test outcomes and symptoms is often weak [28], and our data support this. There was no predictive symptom of persisting nerve damage, although our results may assist future inquiry focusing on impaired touch or temperature sensitivities. Our results demonstrate a heterogeneity of nerve injures following an electrical accident; six of the patients showed signs of small fiber dysfunction only, seven of the patients showed signs of large fiber dysfunction only, and three of the patients showed signs of small and large fiber dysfunction. Thus, from a clinical perspective, detecting underlying functional alterations in this patient group requires a battery of tests to investigate various aspects of nerve function. The implication of these findings shows that although peripheral nerve testing can reveal abnormalities, these tests alone do not make a full clinical evaluation following an electrical accident.

Our cohort was carefully selected and evaluated (Figure 1). We recruited a group of patients that only had symptoms related to their workplace electrical accidents, and not nerve compression or unrelated neuropathy. During this process, there was a substantial drop off. Alongside the fact that our patient group was selected for persisting symptoms above a marginal level, there are factors playing into the various nerve studies. We chose to do a QST with limits because it is fast and a common clinical convention. However, this leaves it vulnerable to reaction times [29]. We saw group-level differences in a number of factors, but only a single clinically relevant instance in LEP. One reason for this is the large variance of the normative data, especially for amplitudes. Using a number of clinically utilized tests of nerve and hand function, we have found results deviating from established norms. This is in contrast to earlier studies concerning this patient group. We believe that this is due to the structure of our study. We recruited a reasonably homogenous group of patients that had had an electrical accident under similar conditions; i.e., mostly low voltage, all entries in the upper extremities, moderate to considerable symptoms persisting for at least a year and controlled for extenuating symptom causes.

We have previously demonstrated altered perceptual abilities following electrical accidents [16], and others have demonstrated altered nerve conduction; for example, in the median nerve [30]. However, few studies have tried to systematically investigate nerve damage as a long-term consequence of an electrical accident. Nerve fibers are at especially high risk of being damaged after an electrical accident because nerves have lower conductance resistance than their surrounding tissues. This can cause nerve-fiber-specific injuries while other tissue types remain unaffected [9]. Therefore, nerve fiber damage may be missed in cases where the overall burn appears small. An animal study suggests that electric shocks cause the most damage in large, fast-conducting, myelinated nerve fibers [31]. Further nerve damage may be due to secondary lesions caused by damage to surrounding tissue and associated swelling and alteration in blood flow [32]. Such indirectly caused dysfunctions of the nerve might show delayed development, from within a few weeks to as much as two years after an accident [33]. Similar delayed consequences have been described for spinal cord function after electrical accidents [34].

Although our patients were carefully selected for accident specific nerve damage, they were not assessed psychologically beyond inquiring into previous or current neurological diagnoses. It is known that electrical accidents can have profound effects on the central nervous system, such as; depression, cognitive deficient, fatigue, motor dysfunction, posttraumatic stress disorder, and more [18,35,36,37]. This could be seen as a limitation of the study, or as a feature of the scope used, namely approaching this investigation with peripheral nerve function in mind. However, using techniques such as LEP have clear central nervous system components. Moreover, one could make an argument that all the measures have central aspects, especially since the information that they contribute has no meaning without the connection between the central and peripheral nervous system. Although we chose to investigate aspects specific to nerve damage, additional criteria could have been relevant to assess persisting symptoms, such as a recent article emphasizing the important of assessing biopsychosocial factors following electrical injury [37]. What is clear is that peripheral nerve evaluations are not the only evaluations critical to post-accident care.

We have demonstrated that electrical accidents can have long-term consequences affecting both motor nerves and small and large diameter sensory nerves. Future studies need to increase the sample size to be able to test for specific associations of the type of injury or bodily location of the accident with nerve dysfunction consequences [38]. Another approach would be a long-term investigation of patients, following their perceptual changes and nerve dysfunctions for several years after the accident. Such a longitudinal study could also contribute evidence to the question of whether secondary compensatory mechanisms are involved. Furthermore, we might explore personalized evaluations for patients being treated after an electrical accident based on the symptoms they present. For example, there is reason to believe that an autonomous nervous function test, such as sympathetic skin response test could predict electric burn victims’ future symptom levels [15]. If evaluations lead to an early diagnosis, nerve damage may even be counteracted, and symptoms mitigated, using various neuromodulator techniques [39,40].

## 5. Conclusions

We have established that a carefully selected electrical accident group with moderate to considerable persisting symptoms can show nerve abnormalities with standard clinical evaluations. We did not establish a clear link between test results and symptoms; a result that confirms the lack of clear patterns between clinical measures and patients’ experiences. To develop better treatment, increase life satisfaction, and well-being following electrical accidents, further studies of the physiological and psychological consequences are necessary.

## Figures and Tables

**Figure 1 brainsci-12-01301-f001:**
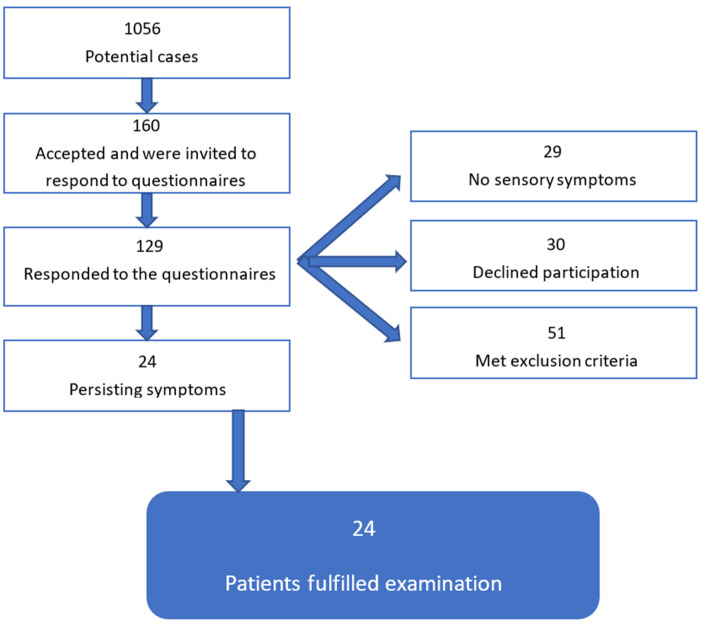
Procedure for inclusion in the study, starting with the 1056 cases provided by the AFA insurance company to the 24 patients tested.

**Figure 2 brainsci-12-01301-f002:**
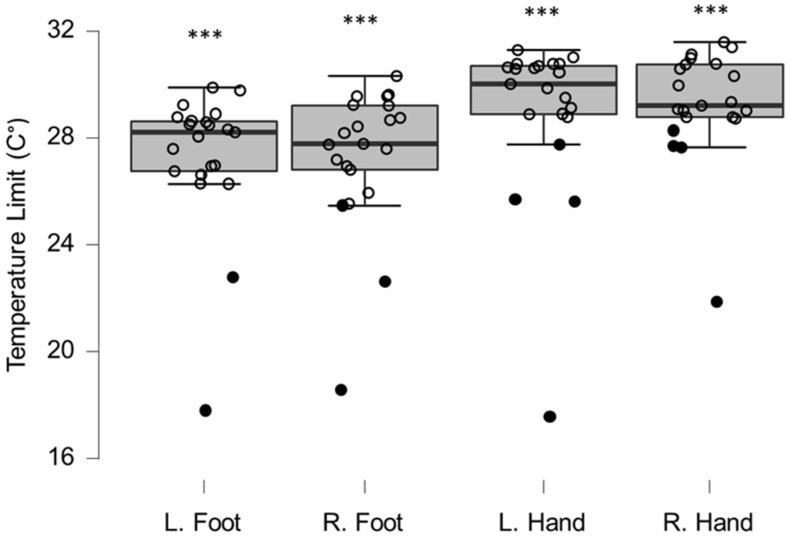
Box-plots for cold sensation limits. Lower and upper box boundaries represent the 25th and 75th percentiles, respectively. The line inside the box represents the median and the lower and upper whiskers represent the 10th and 90th percentiles. Filled circles represent values ≥ two standard deviations outside reference values. *** indicates a statistically significantly lower median values compared to the reference material, with a Bonferroni-corrected *p*-value corresponding to 0.001.

**Figure 3 brainsci-12-01301-f003:**
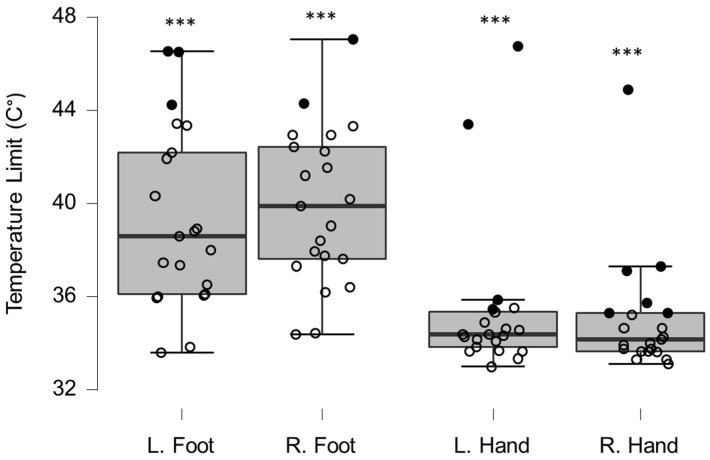
Box-plots for warm sensation limits. Filled circles represent values ≥ two standard deviations outside reference values. *** indicates a statistically significantly lower median values compared to the reference material, with a Bonferroni-corrected *p*-value corresponding to 0.001.

**Figure 4 brainsci-12-01301-f004:**
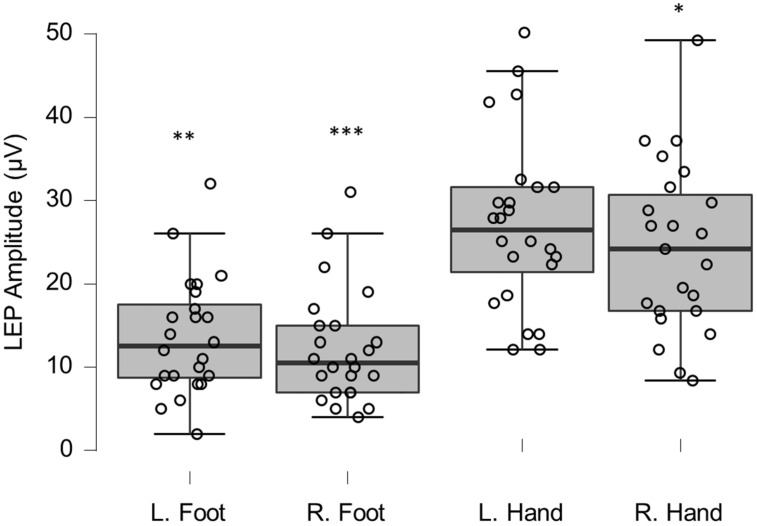
Box-plots for LEP amplitudes. Lower and upper box boundaries represent the 25th and 75th percentiles, respectively. The line inside the box represents the median and the lower and upper whiskers represent the 10th and 90th percentiles. Filled circles represent values ≥ two standard deviations outside reference values. *, **, and *** indicate statistically significantly lower median values compared to the reference material, with Bonferroni-corrected *p*-values corresponding to 0.05, 0.01, and 0.001, respectively.

**Figure 5 brainsci-12-01301-f005:**
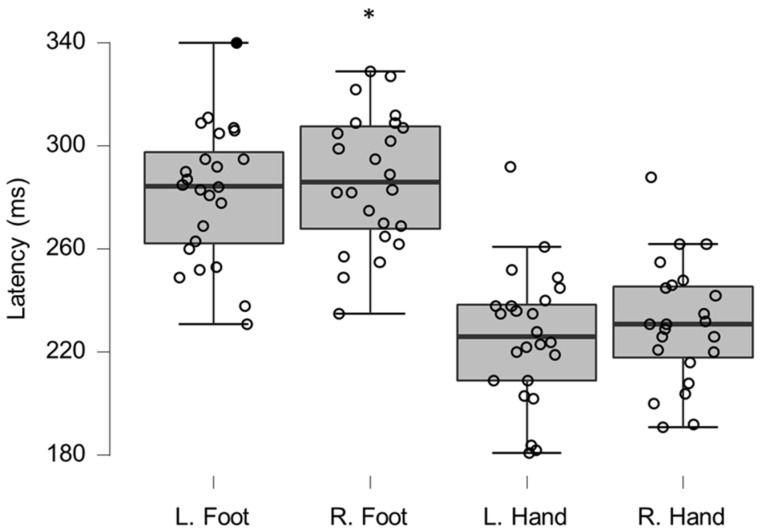
Box-plots for LEP latencies. Filled circles represent values ≥ two standard deviations outside reference values. * indicates a statistically significantly lower median values compared to the reference material, with a Bonferroni-corrected *p*-value corresponding to 0.05.

**Figure 6 brainsci-12-01301-f006:**
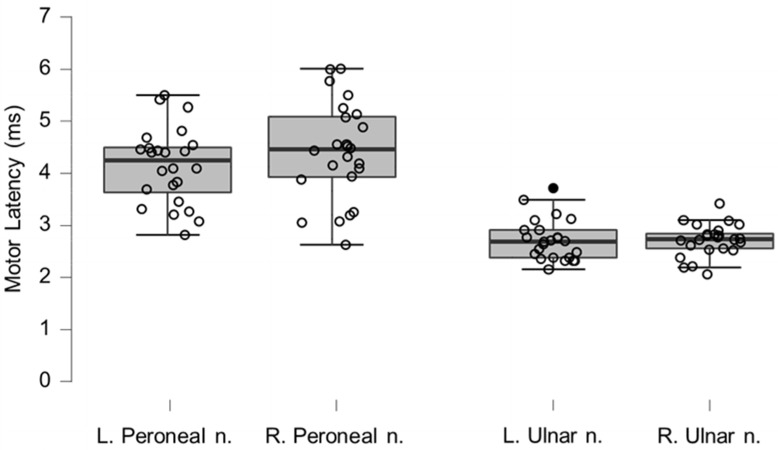
Box-plots for motor neurography latencies. For box plots summary and explanation of symbols, see Figure 2.

**Figure 7 brainsci-12-01301-f007:**
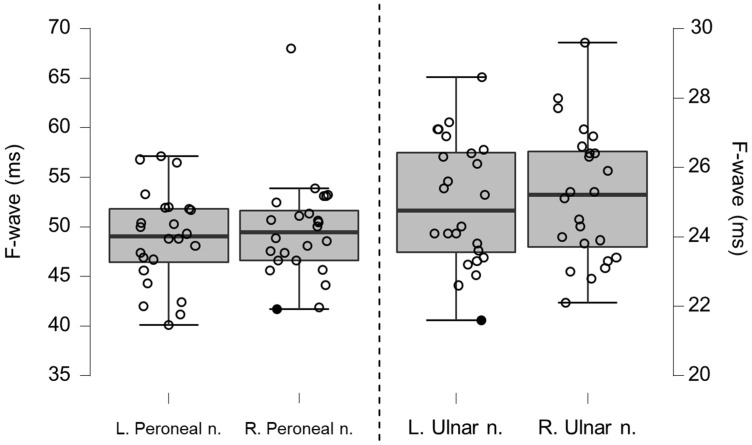
Box-plots for motor neurography F-wave latencies. Filled circles represent values ≥ two standard deviations outside reference values.

**Figure 8 brainsci-12-01301-f008:**
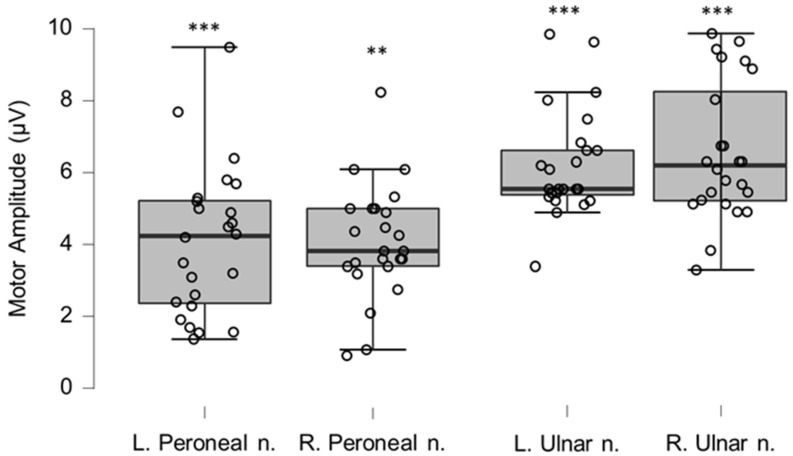
Box-plots for motor neurography amplitudes. For box plots summary and explanation of symbols, see Figure 2. ** and *** indicate statistically significantly lower median values compared to the reference material, with Bonferroni-corrected *p*-values corresponding to 0.01 and 0.001, respectively.

**Figure 9 brainsci-12-01301-f009:**
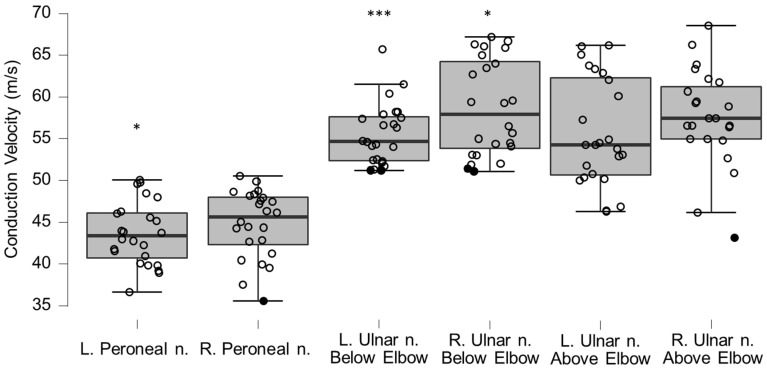
Box-plots for motor neurography conduction velocities. Lower and upper box boundaries represent the 25th and 75th percentiles, respectively. The line inside the box represents the median and the lower and upper whiskers represent the 10th and 90th percentiles. Filled circles represent values ≥ two standard deviations outside reference values. * and *** indicate statistically significantly lower median values compared to the reference material, with Bonferroni-corrected *p*-values corresponding to 0.05 and 0.001, respectively.

**Figure 10 brainsci-12-01301-f010:**
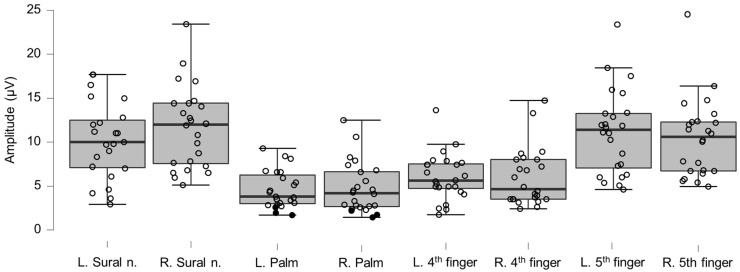
Box-plots for sensory neurography amplitudes. Lower and upper box boundaries represent the 25th and 75th percentiles, respectively. The line inside the box represents the median and the lower and upper whiskers represent the 10th and 90th percentiles. Filled circles represent values ≥ two standard deviations outside reference values.

**Figure 11 brainsci-12-01301-f011:**
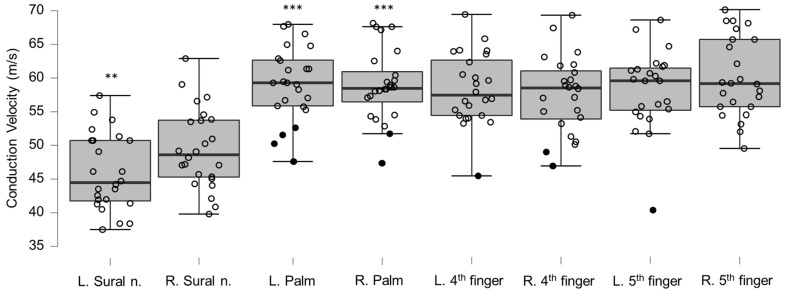
Box-plots for sensory neurography conduction velocities. Lower and upper box boundaries represent the 25th and 75th percentiles, respectively. The line inside the box represents the median and the lower and upper whiskers represent the 10th and 90th percentiles. Filled circles represent values ≥ two standard deviations outside reference values. ** and *** indicate statistically significantly lower median values compared to the reference material, with Bonferroni-corrected *p*-values corresponding to 0.01 and 0.001, respectively.

**Figure 12 brainsci-12-01301-f012:**
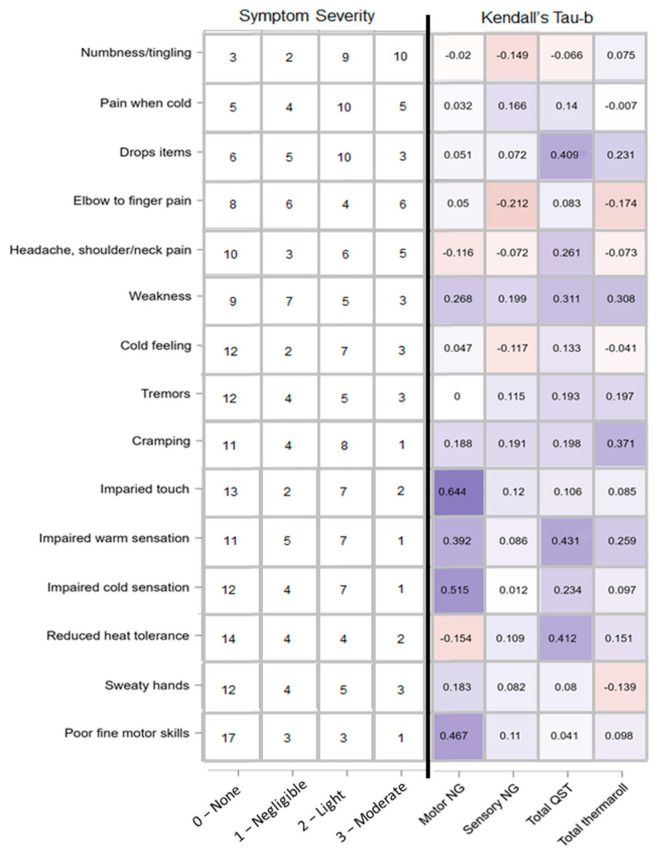
Reported symptom severity and correlation with amount of clinically relevant cases per test paradigm. Left: self-reported, ranked (most prevalent at the top, least at the bottom) symptoms severity adapted from the HAVS questionnaire. Numbers represent number of patients who chose a given intensity option ranging from none to moderate. Right: heatmap of Kendall tau-b correlations representing patients’ self-reported symptoms and amount of clinically relevant cases for test paradigms motor NG (motor neurography), sensory NG, total QST (quantitative sensory testing), and total temperature roller cases.

**Table 1 brainsci-12-01301-t001:** QST limit values for cold and warm sensations on the palm of the hand and arch of the foot. Reference values derived from manufacturers published data from healthy subject normative data. SD is standard deviation. *** indicates a statistically significant difference to reference material with *p*-values corresponding to Bonferroni corrected 0.001.

QST Limits (°C)	Mean	*n*	SD	*p*-Value	
Left hand, cold	30.04	18	0.98	<0.00025	***
Reference hand, cold	31.19	40	0.46		
Right hand, cold	29.64	20	1.24	<0.00025	***
Left foot, cold	28.00	19	1.21	<0.00025	***
Reference foot, cold	30.25	40	1.59		
Right foot, cold	27.74	20	1.87	<0.00025	***
Left hand, warm	34.38	19	0.78	<0.00025	***
Reference hand, warm	32.62	40	0.28		
Right hand, warm	34.50	20	1.19	<0.00025	***
Left foot, warm	39.36	21	3.92	<0.00025	***
Reference foot, warm	34.81	40	2.24		
Right foot, warm	39.97	21	3.43	<0.00025	***

**Table 2 brainsci-12-01301-t002:** LEP amplitudes in μV for evoked potentials on left and right dorsum of the hand or foot. Reference material derived from in-house, healthy subject normative data. *, **, and *** indicate statistically significant differences to reference material with *p*-values corresponding to Bonferroni corrected 0.05, 0.01, and 0.001, respectively.

Amplitude (μV)	Mean	*n*	SD	*p*-Value	
Left hand	24.13	23	9.99	0.0259	
Reference hand	29.05	105	13.37		
Right hand	22.09	23	11.02	0.0061	*
Left foot	13.63	24	7.02	0.0006	**
Reference foot	19.71	87	9.42		
Right foot	11.39	23	5.61	<0.0005	***

**Table 3 brainsci-12-01301-t003:** LEP latencies in milliseconds for evoked potentials on left and right back of the hand or top of the foot. Reference material derived from in-house healthy subject normative data. * indicates a statistically significant differences to reference material with *p*-values corresponding to Bonferroni corrected 0.05.

Latency (ms)	Mean	*n*	SD	*p*-Value	
Left hand	223.26	23	22.25	0.1185	
Reference hand	227.19	105	19.93		
Right hand	230.87	23	23.77	0.2475	
Left foot	281.79	24	26.05	0.1088	
Reference foot	274.08	87	28.89		
Right foot	287.04	24	25.83	0.0189	*

**Table 4 brainsci-12-01301-t004:** Motor neurography latencies for left and right peroneal and ulnar nerves.

Motor Neurography				
Latency (ms)	Mean	*n*	SD	*p*-Value
Left peroneal nerve	4.42	24	0.94	0.7147
Reference peroneal nerve	4.54	54	0.74	
Right peroneal nerve	4.15	24	0.73	0.9830
Left ulnar nerve	2.71	24	0.39	0.9821
Reference ulnar nerve	2.90	100	0.30	
Right ulnar nerve	2.69	23	0.49	0.9926

**Table 5 brainsci-12-01301-t005:** Motor neurography F-wave latencies for left and right peroneal and ulnar nerves.

F-Wave (ms)	Mean	*n*	SD	*p*-Value
Left peroneal nerve	48.10	23	4.94	0.9248
Reference peroneal nerve	46.77	54	2.43	
Right peroneal nerve	47.06	23	3.79	0.6639
Left ulnar nerve	24.98	24	1.81	0.5949
Reference ulnar nerve	24.89	100	1.09	
Right ulnar nerve	25.20	24	1.92	0.7771

**Table 6 brainsci-12-01301-t006:** Motor neurography amplitudes for left and right peroneal and ulnar nerve. ** and *** indicate statistically significant differences to reference material with *p*-values corresponding to Bonferroni corrected 0.01, and 0.001, respectively.

Amplitude (μV)	Mean	*n*	SD	*p*-Value	
Left peroneal nerve	4.1	24	2.07	0.0003	***
Reference peroneal nerve	6.08	54	2.46		
Right peroneal nerve	4.48	22	1.12	0.0005	**
Left ulnar nerve	6.30	22	1.17	<0.0005	***
Reference ulnar nerve	10.1	100	3.30		
Right ulnar nerve	6.83	24	1.77	<0.0005	***

**Table 7 brainsci-12-01301-t007:** Motor neurography conduction velocities (CV) for left and right peroneal and ulnar nerves. * and *** indicate statistically significant differences to reference material with *p*-values corresponding to Bonferroni corrected 0.05 and 0.001, respectively.

Conduction Velocity (m/s)	Mean	*n*	SD	*p*-Value	
Left peroneal nerve	43.57	24	3.76	0.0061	*
Reference peroneal nerve	45.96	54	3.68		
Right peroneal nerve	44.73	24	4.05	0.1049	
Left ulnar nerve, below elbow	55.44	24	3.67	0.0008	***
Reference ulnar nerve	61.60	100	5.30		
Right ulnar nerve, below elbow	58.68	24	5.75	0.015	*
Left ulnar nerve, above elbow	56.45	24	7.07	0.5905	
Reference ulnar nerve	56.10	100	5.20		
Right ulnar nerve, above elbow	58.10	22	5.09	0.9209	

**Table 8 brainsci-12-01301-t008:** Sensory neurography amplitudes for left and right sural and ulnar nerves.

Sensory Neurography				
Amplitude (μV)	Mean	*n*	SD	*p*-Value
Left sural nerve	11.04	23	4.38	1
Reference sural nerve	2.41	165	0.58	
Right sural nerve	10.53	24	5.02	1
Left ulnar palm	4.49	24	2.18	0.9863
Reference ulnar palm	3.44	89	0.64	
Right ulnar palm	4.81	24	2.89	0.9848
Left ulnar 4th finger	5.65	23	2.17	1
Reference ulnar 4th finger	2.01	89	0.49	
Right ulnar 4th finger	5.96	24	3.25	1
Left ulnar 5th finger	10.06	24	5.39	1
Reference ulnar 5th finger	2.15	89	0.55	
Right ulnar 5th finger	8.53	23	3.67	1

**Table 9 brainsci-12-01301-t009:** Sensory neurography conduction velocities for left and right sural and ulnar nerves. ** and *** indicate statistically significant differences to reference material with *p*-values corresponding to Bonferroni corrected 0.01, and 0.001, respectively.

Conduction Velocity (m/s)	Mean	*n*	SD	*p*-Value	
Left sural nerve	45.96	24	5.66	0.0006	**
Reference sural nerve	50.43	165	6.03		
Right sural nerve	49.40	24	5.94	0.2176	
Left ulnar palm	59.13	24	5.34	0.000167	***
Reference ulnar palm	69.50	89	8.54		
Right ulnar palm	58.91	24	5.29	0.000167	***
Left ulnar 4th finger	58.23	24	5.27	0.0303	
Reference ulnar 4th finger	60.61	89	5.66		
Right ulnar 4th finger	57.46	24	5.68	0.0104	
Left ulnar 5th finger	58.41	21	4.13	0.3385	
Reference ulnar 5th finger	60.10	89	5.81		
Right ulnar 5th finger	60.00	24	5.93	0.4692

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
