# Peer review of "Clinical Evaluation of Nerve Function in Electrical Accident Survivors with Persisting Neurosensory Symptoms"

_brainsci, 2022, doi:10.3390/brainsci12101301_

Round 1
Reviewer 1 Report
Thank you an interesting study. I find it quite fascinating and important.
first you have to add the demographics of the patients. EI, location, time off, rehab, what mean persistent problems, mental health etc all of this needs to be presented.
what is the consequence of your findings?
many and larger trials showed before NO change at all can be detected. why can you find a difference all for sudden?
the paper needs some better focus, hypothesis and aim.
Author Response
Thank you an interesting study. I find it quite fascinating and important.
We appreciate it, thank you.
first you have to add the demographics of the patients. EI, location, time off, rehab, what mean persistent problems, mental health etc all of this needs to be presented.
We expanded section 3.1 to include more demographic information and even included a supplementary table with more information about the patients. We also added an explanation in the introduction about persisting symptoms: “…and the patient’s persisting symptoms, defined as symptoms present more than one-year after an accident.” Finally, we added a limitations section in the discussion to address the lack of psychological evaluations (since the focus was on peripheral nerve damage).
what is the consequence of your findings?
We thank you for showing us that this was not clear enough in the manuscript. Now we have added: “We establish that a carefully selected electrical accident group with moderate to considerable persisting symptoms can show nerve abnormalities with standard clinical evaluations.”
many and larger trials showed before NO change at all can be detected. why can you find a difference all for sudden?
We believe this has to do with our selected group having peripheral persisting symptoms at a moderate to considerate level as an inclusion and otherwise homogenous categorization. This argument is given in the discussion, paragraph four. We have expanded it to accommodate your question:
“Our cohort was carefully selected and evaluated (Figure 1). We recruited a group of patients that only had symptoms related to their workplace electrical accidents, and not nerve compression or unrelated neuropathy. During this process, there was a substantial drop off. Alongside the fact that our patient group was selected for persisting symptoms above a marginal level, there are factors playing into the various nerve studies. We chose to do a QST with limits because it is fast and a common clinical convention. However, this leaves it vulnerable to reaction times 29. We saw group-level differences in a number of factors, but only a single clinically relevant instance in LEP. One reason for this is the large variance of the normative data, especially for amplitudes. Using a number of clinically utilized tests of nerve and hand function, we have found results deviating from established norms. This is in contrast to earlier studies concerning this patient group. We believe that this is due to the structure of our study. We recruited a reasonably homogenous group of patients that had had an electrical accident under similar conditions; i.e., mostly low voltage, all entries in the upper extremities, moderate to considerable symptoms persisting for at least a year and controlled for extenuating symptom causes.”
the paper needs some better focus, hypothesis and aim.
Yes, this is understandable and thank you for identifying it. To address this, we’ve changed the second to last paragraph of the introduction to frame our scope better, with passages like: “Our focus is on peripheral nerve function, … following electrical accidents” and to make the aim clears with “Our aims are to test if clinical routines are able to detect nerve function abnormalities within a highly selected group of electrical accident patients with persisting symptoms related to their accident, and, if such abnormalities can be found, to establish which persisting symptoms best predict subsequent nerve function.”
The last paragraph of the introduction now explicitly states our hypotheses: “Knowledge is scarce regarding peripheral nerve function after electrical accidents 17. We hypothesized that in a carefully selected group of patients displaying adequate long-term symptoms at least one abnormal neurophysiological test value will be present. Treating acute electrical accidents focuses on immediate care, while neurophysiological symptoms often appear weeks to months after an accident 18. Therefore, we only included a group selected for symptoms that have persisted for at least a year, and, to increase the chance of having an abnormal test result, only include those with symptoms of a moderate to considerable severity level. We also hypothesized that there would be a link between the tests with abnormal results and the persisting symptoms that resulted from an electrical accident.”
Reviewer 2 Report
In essence, with minor revisions. The focus of the study was on the peripheral nervous system, whereas it is my view, as a pain medicine physician, that the central nervous system is critically involved and that should be acknowledged for balance. The authors were apparently unaware of the recent article, that Prof Chris J Andrews (co-author) is an authority on electrical injuries.
The authors did a good job of testing peripheral nerve function but totally ignored dysfunction in the central nervous system which was shown to be important in our publication.
Please see attachment for additional comments. 9/5/2022

Author Response
In essence, with minor revisions. The focus of the study was on the peripheral nervous system, whereas it is my view, as a pain medicine physician, that the central nervous system is critically involved and that should be acknowledged for balance. The authors were apparently unaware of the recent article, that Prof Chris J Andrews (co-author) is an authority on electrical injuries.
Thank you for the resource and your feedback. We have added more about the CNS in the discussion and added a number of references include the 2022 Injury publication in the revised manuscript. We also agree that addressing the CNS is important to balance the discussion of most relevant factors following an electrical accident. That paragraph in the discussion now look like this (see manuscript for references):
“Although our patients were carefully selected for accident specific nerve damage, they were not assessed psychologically beyond inquiring into previous or current neurological diagnoses. It is known that electrical accidents can have profound effects on the central nervous system, such as; depression, cognitive deficient, fatigue, motor dysfunction, posttraumatic stress disorder, and more 18, 35, 36, 37. This could be seen as a limitation of the study, or as a feature of the scope used, namely approaching this investigation with peripheral nerve function in mind. However, using techniques such as LEP have clear central nervous system components. Moreover, one could make an argument that all the measures have central aspects, especially since the information that they contribute has no meaning without the connection between the central and peripheral nervous system. Although we chose to investigate aspects specific to nerve damage, additional criteria could have been relevant to assess persisting symptoms, such as a recent article emphasizing the important of assessing biopsychosocial factors following electrical injury 37. What is clear is that peripheral nerve evaluations are not the only evaluations critical to post accident care.
The authors did a good job of testing peripheral nerve function but totally ignored dysfunction in the central nervous system which was shown to be important in our publication.
Thank you, I should say that it was not intentional to ignore the CNS, it was to emphasis that this study’s aim was to investigate the peripheral nervous system, specifically. The use of LEP and even the other measurements can provide some insight into the CNS, but we hypothesized that taking the PNS focus may give us different insights into how clinical routines can be used to identify nerve abnormalities and how to possible improve those routines. We think the data shows some tests are able to provide some insights, but this battery of tests is clearly not comprehensive in assessing all symptoms following an electrical accident.
We added passages that makes the focus on the PNS clearer, expanded the discussion to include the CNS, and included a reference to previous work on electrical injury and the CNS (including Lun et al. 2022).
Reviewer 3 Report
Do not start sentences with numbers and acronyms, e.g., line 18: 67% of the patients…, also in lines 76, 79, … check the whole paper for that.
Explain in more detail the implications of “No test result strongly correlated with self-reported symptoms”.
The sentence in Line 41 could use a citation.
Two periods in Line 57.
Second paragraph in section 2.1: you first report the number 12 and only then explain what the test represent, while neglecting to mention what the number 12 means comparatively. Try to introduce/explain the test before reporting the results, and then explain the results; not the other way around.
The data collection is comprehensive and well processed. Compliment on including the box-plots. Many researchers in similar fields only present tables with numbers that can be difficult to read.
The conclusion section is well written, too.
Congratulations on study well done and manuscript well written.
Finish your ‘authors contributions’ section.
Author Response
Do not start sentences with numbers and acronyms, e.g., line 18: 67% of the patients…, also in lines 76, 79, … check the whole paper for that.
Thanks, these aspects have been fixed throughout the manuscript.
Explain in more detail the implications of “No test result strongly correlated with self-reported symptoms”.
This is a difficult point to make, and some may think it is not particularly important. The interesting finding here is the lack of a clear relationship between test outcomes and symptoms. It is a challenge to discuss something that is not there and even harder to speculate why it is not, but we attempted to expand our explanation in the manuscript with the following:
[In the 3rd paragraph of the discussion]: “There was no symptom that stood out as particularly predictive of persisting nerve damage, although our results may assist future inquiry focusing on impaired touch or temperature sensitivities.”
[In the summary paragraph]: “There was only weak correlation between symptoms and test outcomes; a result that confirms the lack of clear patterns between clinical measures and patients’ experiences.”
The sentence in Line 41 could use a citation.
Good point, it has been added.
Two periods in Line 57.
Well spotted, thanks.
Second paragraph in section 2.1: you first report the number 12 and only then explain what the test represent, while neglecting to mention what the number 12 means comparatively. Try to introduce/explain the test before reporting the results, and then explain the results; not the other way around.
Thank you for this insight, it should be clearer. We changed the order and expanded the explanation in the manuscript. This is the new version:
“A total of 129 individuals (81%) responded to the initial questionnaire. Besides having had an electrical accident, we needed to ensure that participants had persisting symptoms that pertained to their accident and not to another source. Our main inclusion criteria utilized two tests to verify electrical accident persisting symptoms: Self-Administrated Leeds Assessment of Neuropathic Symptoms and Signs (SLANSS) and the hand-arm vibration syndrome (HAVS) scale. SLANSS is a validated 11-item scale used to uncover the presence of neuropathic pain 17. Participants can have a max score of 24, and a score of 12 or above indicates a presence of moderate to considerable symptoms. The HAVS is a validated questionnaire 10, 16 that contains 10-items about hand symptoms with the scaled options of no, light, moderate and considerable. To be included, participants had to have a SLANSS score of ≥12 and report moderate or considerable symptoms on HAVS. “
The data collection is comprehensive and well processed. Compliment on including the box-plots. Many researchers in similar fields only present tables with numbers that can be difficult to read.
Thank you, we think the plots add more depth than just tables. We are glad you appreciate them.
The conclusion section is well written, too.
Thank you.
Congratulations on study well done and manuscript well written.
Also, much appreciated.
Finish your ‘authors contributions’ section.
It had been added to the manuscript. Thanks for the reminder.